# Incorporating Rarity and Accessibility Factors into the Cultural Ecosystem Services Assessment in Mountainous Areas: A Case Study in the Upper Reaches of the Minjiang River

**Yafeng Lu** [1] , **Qinwen Li** [1,2] , **Pei Xu** [1] **and Yukuan Wang** [1,*]

[1]   Institute of Mountain Hazards and Environment, Chinese Academy of Sciences and Ministry of Water Resources, Chengdu 610041, China; luyafeng@imde.ac.cn (Y.L.); lesleyluvlee@gmail.com (Q.L.); xupei@imde.ac.cn (P.X.)

[2]   University of Chinese Academy of Sciences, Beijing 100049, China

*   Correspondence: wangyukuan@imde.ac.cn

**Abstract:** Cultural ecosystem services (CES) are not only a key source for supporting the development of economy but also maintain the ecological security in mountainous areas. However, there are limited numbers of studies that focus on establishing the assessment model for the CES at a regional scale. We combined the topographic factors and accessibility factors to quantify the distribution of CES and tested the approach with data on road and topography in the upper reaches of the Minjiang River. The results showed that the areas with high CES were located in the southwestern part of the study area, where it was convenient traffic and rare topography. Results from our approach were likely to support the development of local tourism industry because the distribution of CES was consistent with current hotspots for scenic spots. Meanwhile, we found that the area with high rarity and low accessibility should improve accessibility in order to enhance the capacity of CES. The assumptions applied in our approach highlighted the impacts of complex topography on CES, which could be suitable for the area with a lack of data. Moreover, our approach provided an effective way to assess CES for creating management strategies and enhancing capacity in mountainous areas.

**Keywords:** ecosystem services; cumulative viewshed analysis; K-means clustering algorithm; tourism industry; management

## 1. Introduction

Cultural ecosystem services (CES) are the key sources that support production and livelihood in mountainous areas [1]. With the development of urbanization, a rural revitalization becomes a long-standing challenge and a top priority for the local government in mountainous areas [2,3]. Most scientists and managers agree that the development of tourism industry could boost economic development and improve livelihood, which is a preferred strategy based on the ecological sources and local culture in mountainous areas [4,5]. CES are considered the basis for the tourism industry, while maintaining ecological security and conserving biodiversity also depended on sustainable use of ecosystem services. The comprehensive understanding of the distributions and values of CES played a significant role in the management for natural resources and rural development. In general, CES not only represent purely ecological sources but are the outcome of coupling relationships between humans and lands in landscapes [6]. Current thinking increasingly recognizes the need to integrate natural factors and human preferences into the CES assessment [7]. Despite this, there are still limited models to demonstrate how CES should be calculated at a regional scale. As a result, establishing the

assessment approach is a significant prerequisite for ecosystem service management and conservation in mountainous areas.

During the last decades, methods of assessing CES have developed from individual investigation to economic analysis, and are divided into four approaches, including visitor expenditures analysis [8], assessing accessibility of habitats [9,10], mapping benefits in terms of transfer of monetary [11] and stakeholders' investigation [12]. Recent advances in assessing CES have extended spatial analysis methods, participatory methods, including viewshed analysis, analytic hierarchy process [13] and hotspots identification [14]. However, the current approach focused on the impacts of subject on CES rather than the effects of ecological structures and functions, while previous studies have been limited to explore the relationships between ES and the demand for society or human preferences [3]. Meanwhile, results from these models also demonstrated a large bias [8].

The solution lies in the understanding of CES produced by ecosystems as well as were influenced by social preferences in a region. Generally, changes of ecosystems determined the value of CES and were related to the human experience and appreciation of CES [15]. Previous studies suggested that landscape patches contributed to the value of CES, while assessment patches differ, including composition, structure and ecological maturity, could quantify landscape value [16]. In mountain areas, complex topography and geomorphology provided richness habitats, which shaped the rare landscape and local culture. Quantifying the effects of topography could identify the spatial differences of CES in terms of changes in topography in a landscape. Meanwhile, although the topography enhanced the capacity of CES, it influenced of accessibility on CES. Therefore, a reasonable approach should incorporate the accessibility and rarity into the assessment of CES.

Owing to the effects of topography on CES in mountainous areas, we develop the assessment model to quantify accessibility and rarity for assessing CES. Specifically, (1) assessing the accessibility by cumulative viewshed analysis algorithm, (2) identifying the rarity in each viewshed, (3) incorporating accessibility and rarity into the CES assessment.

## 2. Methodology

### 2.1. Study Area

The upper reaches of the Minjiang River is the eastern part of the Tibet plateau located in the Sichuan province of western China (Figure 1). It covers an area of 24,121 km$^2$ and includes five counties: Wenchuan, Maoxian, Lixian, Heishui and Songpan. It is not only a significant area economically but also a hotspot for ecological conservation in southwestern China. There are many national protection species in this area, including giant panda (*Ailuropoda melanoleuca*). For a long time, ecosystem services in this area were vital for agricultural and industrial production [17]. Recently, tourism has gradually become the key industry, which dramatically promotes the economic development of this area [18]. However, unreasonable development plans and exploitation may result in degradation and destruction of ecosystems and their services [19]. Thus, quantifying assessment the CES could guide the economic development, sustainable management and biodiversity of conservation efforts.

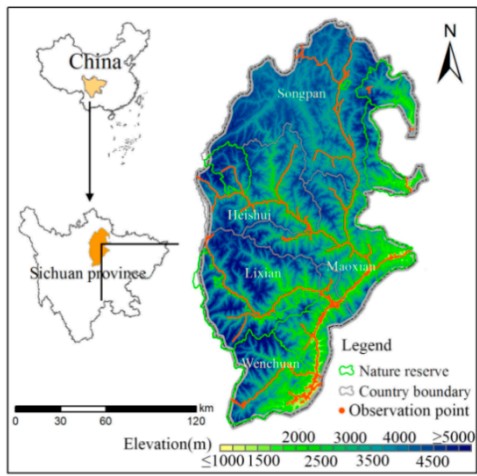

**Figure 1.** Location of the study area.

## 2.2. Methodology

In mountainous areas, the distribution of roads determined the accessibility of service. For the rarity, complex topography provide various and rare habitats, which shaped the specific landscapes. Therefore, we incorporated these factors into the CES assessment, and the function is expressed in Equation (1).

$$ES = A \times R \tag{1}$$

where ES represents the CES value, A represents the accessibility, and R represents the rarity in each viewshed.

## 2.3. Assessment Framework

According to the Equation (1), the assessment framework was designed, including three steps in Figure 2.

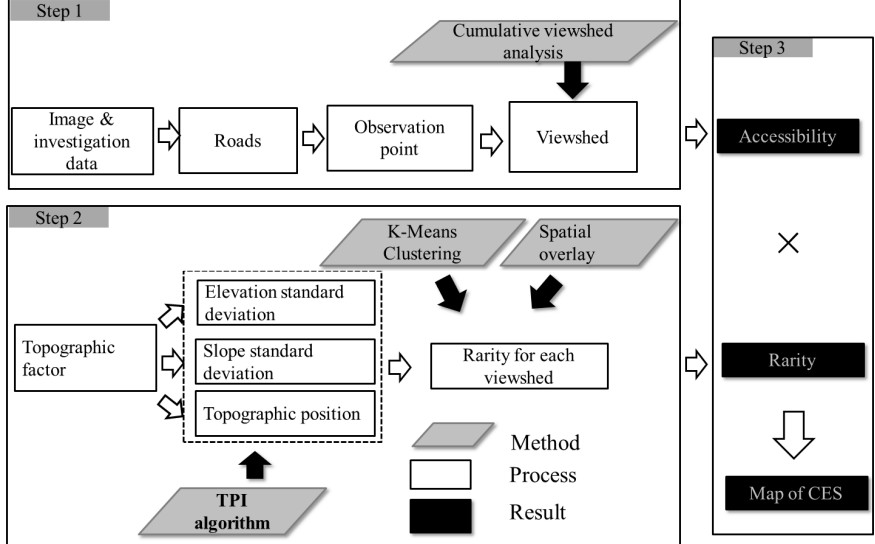

**Figure 2.** The methodological framework proposed to assess cultural ecosystem services (CES).

*2.4. Models*

(1) Cumulative Viewshed Analysis

The cumulative viewshed analysis can be used to make inferences about the relationships among observation points and ecosystems in a region [20], which could demonstrate how the accessibility distributed in the evaluation area. We used the viewshed tool in QGIS (version 3.4, QGIS Development Team, 2012, QGIS Geospatial Graphic Information System, Open Source Geospatial Foundation Project) to analyze the viewshed for each observation points. Then, the map of accessibility was calculated by overlaying analysis.

(2) Rarity

The rarity is characterized by the structures and changes of ecosystem types in the evaluation area. In general, the area with a high rarity could provide more CES. We used a K-means clustering algorithm to quantitatively describe the rarity for topographic conditions in each viewshed, indicating the changes of CES. The percentage of types could be obtained from results of K-means clustering algorithm, which represented different rarities. A high percentage represents a low rarity and vice versa, as shown in Equation (2). Three indices, including elevation standard deviation (SD), slope SD and topography position in each viewshed, were used to calculate the rarity by K-means clustering algorithm.

$$R = 1 - (\frac{\sum n_k}{N} \times 100\%) \tag{2}$$

where R represents the rarity in each viewshed, N represents the number of viewsheds, and nk represents the number of viewsheds of the k cluster, and k is the number of clusters from K-Means clustering algorithm.

The K-means clustering algorithm is a widely used partitioning algorithm that can divide data into K mutually exclusive clusters. By using the K-means algorithm, variables from each viewshed as an object have different locations and find a partition in which objects within each cluster are as close to each other as possible, and as far from objects in other clusters as possible. The number of clusters was divided by the iterative algorithm that minimizes the sum of Euclidean distance from each object to its centroid of the cluster.

Additionally, Wiess suggested that topographic position index (TPI) was an effective method to identify topographic positions by comparing the elevation value of each cell in a region to the value of a specified neighborhood around that cell [21]. In TPI, positive values suggested that the elevation in the cell is higher than that in surrounding cells, which was identified as ridges. Similarly, negative TPI values and TPI near-zero were defined as valley and flat areas, respectively. In order to obtain a more reasonable result, a distance function was used to improve the TPI algorithm, as shown in Equation (3).

$$TPI_{new} = \frac{1}{n} \sum_{i=1}^{n} \frac{TPI_i}{d_i} \tag{3}$$

where TPInew is improved TPI, n represents the radius of sliding windows (n = 15), which were used to calculate the relationship between the elevation in the cell and that in surrounding cells. TPIi is the i-th topographic position index based on the i-th radius, and di is distance values for calculating TPI in the i-th radius between target cells and other cells.

According to TPI, the topographic were divided into 11 types, including steep slope—North/North East (N/NE), steep slope—South/South West (S/SW), slope crest, upper slope, flat summit/ridge, sideslope—N/NE, cove/ravine—N/NE, sideslope—S/SW, cove/ravine—S/SW, slope bottom, water.

### 2.5. Data

The CES assessment is closely related to natural conditions and social factors, which mainly involve datasets about basic geographic data, road net, scenic areas and others. Basic geographic data were derived from the digital elevation model (DEM) provided by USGS. The resolution of the DEM was resampled to a raster of 100 m. The TPI index, slope SD and elevation SD were calculated by this DEM. The roads were obtained from the land use data that interpreted from Google Earth of 10-m resolution in 2018. Additionally, the distributions of scenic areas were obtained from local governments, which were verified by a field survey.

### 2.6. Parameters

(1) Observation Point

A large number of observation points ensured a comprehensive and spatially dense coverage of the upper of Minjiang River. In general, most tourists enjoyed CES along the roads in the evaluation area. Thus, a total of 3374 observation points were identified in a 1000 m interval for all roads (Figure 1).

(2) The Number of Clusters

The K-mean algorithm need to determine the K value, which represented initial centroids. Then, an iterative process optimized the centroids to classify it to the closest selected centroid. According to previous studies, the within-cluster sum of squared errors (SSE) and the silhouette coefficient method were used to determine the K value [22,23]. As shown in Figure 3, the K value was set to five clusters.

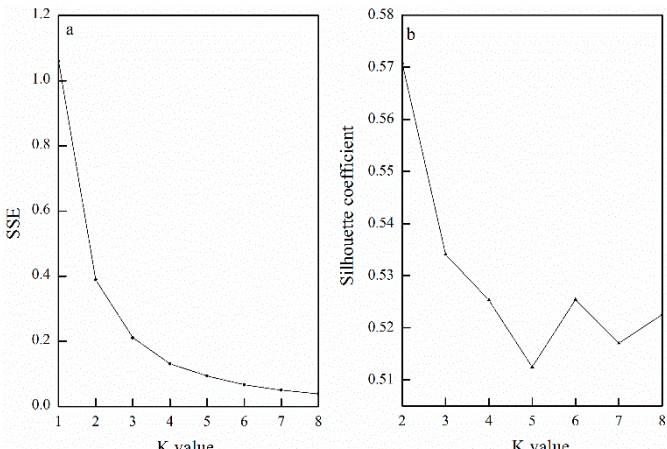

**Figure 3.** Method for determining the K value. The lower the values as calculated by the within-cluster sum of squared errors and the silhouette coefficient, the better the clustering quality of K-mean algorithm.

## 3. Results

### 3.1. Cumulative Viewshed

Applying the viewshed tool in GIS identified the distribution of viewshed for each observation point in the mountainous area (Figure 4a). Results from the cumulative viewshed map showed the total number of areas that were accessible for the ecosystem service across the evaluation area from all of the observation points. In details, the greater value of 25 for cumulative viewshed was mainly distributed in northern and southwestern parts of the study area, where convenient transportation is a vital basis for CES (Figure 4b).

More importantly, results from cumulative viewshed analysis also show that most areas could not provide ecosystem services (account for 72.4% of the study area). These areas may have high ecosystem services, while could not be accessible because of fewer roads. Thus, to assess CES, firstly

divided the study area into two areas, including the areas with provisioning services and the areas with nonprovisioning services.

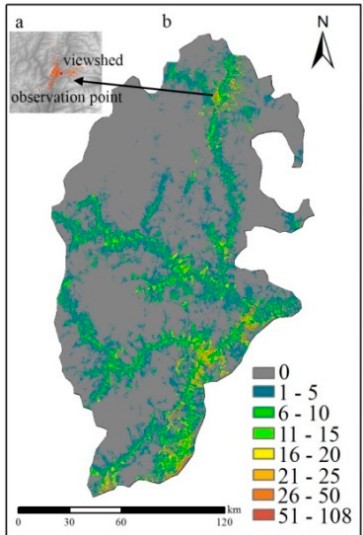

**Figure 4.** Diagram of viewshed analysis for an observation point (**a**) and the distribution of cumulative viewshed (**b**).

## 3.2. The Distribution of Rarity

As shown in Figure 5a, a total of 11 topographic positions were identified in the upper reaches of the Minjiang River by TPI algorithm. The steep slope (N/NE and S/SW) comprised the most area (account for 32.7% of area) and mainly distributed in the southern and eastern part of the study area. Spatially, the degree of topographic diversity decreased with the increase of elevation (Figures 1 and 5a). As the close relationship between topography and rarity, results from K-means clustering analysis showed that moderately degree of rarity prevailed over the most area in this region. While we found that the values for the rarities ranged from 0.70 to 0.92 and were spatially heterogeneous (Figure 5b).

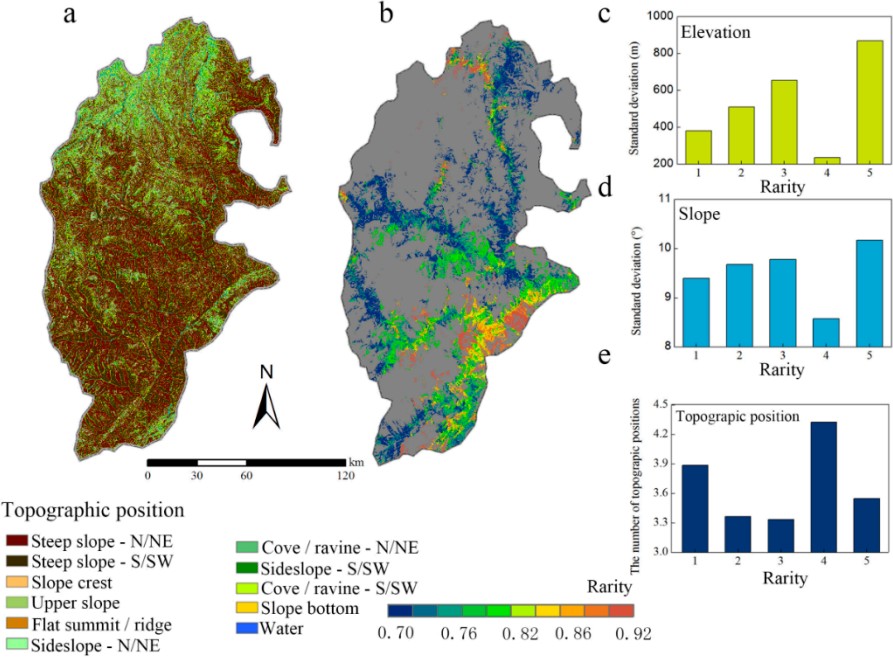

**Figure 5.** The topographic position (**a**), the rarity (**b**) and the relationship among rarity, elevation SD (**c**), Slope SD (**d**), and the topographic position (**e**) in the study area.

Further analyses revealed that the elevation SD and the slope SD, and the topographic positions played different roles in assessing rarity. Results from K-means clustering algorithm showed the degree of rarity increased with the increase of the elevation SD and the slope SD (Figure 5c,d), indicating that greatly undulate terrain in viewsheds are rare formations in this region. Additionally, the topographic position was another key factor for rarity assessment. When the number of topographic positions was more than four types in a viewshed, the rarity can also reach a high level. Thus, incorporating different topographic indices into the rarity assessment could be necessary.

### 3.3. CES in the Upper of Minjiang River

Similarly, the distribution of CES was consistent with the result of cumulative viewshed analysis. In general, the CES value increased with the intensity of the road and from north to south. As shown in Figure 6a, the highest value of CES reached a value of 100 and was located in the Wenchuan county because of convenient traffic and rare topography. However, the area with low values of CES prevailed over the most area in the upper reaches of the Minjiang River.

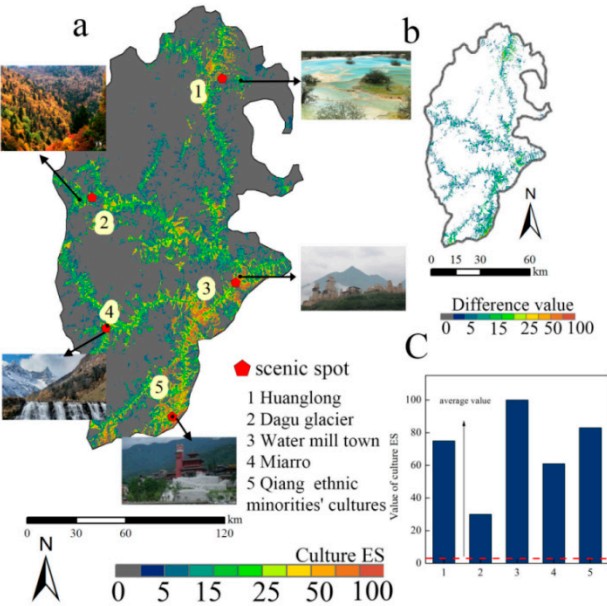

**Figure 6.** The distribution of CES: (**a**) the difference between results from cumulative viewshed analysis and CES assessment, and (**b**) the CES value in the scenic spots (**c**).

Additionally, the differences between results from the CES assessment and results from cumulative analysis suggested the degrees of rarity were sufficient to cause the change in CES (Figure 6b). Furthermore, the Wilcoxon test ($P < 0.05$) showed that there was a significant difference between the Figures 4b and 6a.

To validate our approach, we extracted the value of CES in scenic spots from the assessment result. As shown in Figure 6c, the values of CES in scenic spots were dramatically greater than the average value in the whole evaluation area, indicating that the results were reasonable.

## 4. Discussion

### 4.1. The Impacts of Roads on CES

Our approach suggested that the accessibility and the rarity are key factors of CES in mountain areas. The current distribution of CES is likely shaped by roads and the rare topography. Rather than assessing the hotspot areas for CES [24], our approach is different in its focus on the effects of complex topography on CES in mountainous areas. Additionally, unlike other ecosystem services—such as

water retention—CES is difficult to transfer by natural paths. Thus, accessibility directly influenced services that can be accessed (or not) by human.

Applying cumulative viewshed analysis could describe the spatial distributions of accessibility in mountain areas. First, all observation points from road determined where CES could be accessible in the study area. Previous studies suggested that roads have been determined as one key factor that assessed and mapped CES [13,24]. Additionally, viewshed analysis for each point and the spatial overlay analysis could quantify accessibility, as well as identify hotspots for CES in the evaluation area. Because the number of visits to an area could significantly contribute to the spatial differences in CES [25,26]. Thus, as for the CES assessment, the areas with nonprovisioning services should be identified, as well as explicit quantifications of accessibility.

## 4.2. The Effects of Complex Topography

As the services are produced by ecosystems, the structures and distributions of ecosystems are the core of assessing ecosystem services [15]. In previous studies, topographic factors have been shown to be a good indicator of the distribution of ecosystems in mountainous areas [27]. First, an area with complex topography such as that found in this study likely renders high-level CES. Secondly, a high topographic complexity likely offered to provide more habitats, thus increasing the number of species and ecosystem types as well as possibly also increasing rarity [28]. Thirdly, the low influence of human activities on the ecosystem in areas with complex topography might indicate that the situ species and ecosystem could reserve to maintain a high level of ecosystem services.

Moreover, the local culture and the demand for social preferences are important for CES, which also depend on the topographic factors in mountainous areas. Most previous studies revealed that the positive effect of the presence of water bodies, high mountain and on the cultural ecosystem [29,30]. In our approach, identifying topographic positions and diversity are complementary to culture assessment. Meanwhile, according to our results, the higher the value of CES value in No. 3 and No. 5 scenic spots, to some extent, (Figure 6a) suggested that the area with specific topography could stand for the local culture and preferences. In addition, the hot spots for tourism in the upper reaches of the Minjiang River changed along elevational and geomorphological characteristics. In the relatively low elevation belts, linkages with unique culture and ecosystems have already been identified as the advantages of CES. In higher elevation regions, the variation of topography shaped the richness and diversity of ecosystems, which provide favorable conditions for tourists, such as No.1 scenic spots in Figure 6a. Thus, besides topographic positions, both elevation SD and slope SD are two critical factors for attention.

Furthermore, although most several previous studies highlighted the rarity should be considered in the CES assessment because it was close the relationship between CES and tourism [31], these approaches do not establish the approach of rarity assessment. In most studies, the location of digital photos was used to represent the rarity in a region [24,32]. Another effective approach explored individual preferences and calculated ecosystem services by interviews or questionnaires [33]. These approaches could reveal the relationship between social factors and ecosystem service, while may accurately calculate the hotspots in CES. However, the distributions and structures of ecosystems were ignored, thus, these assessing approach may be suitable for scenic spots in a small area. Comparatively, the simplicity of a rarity assessment based on the K-means clustering algorithm makes our approach more general and is also easily understood. This approach is suitable for assessing CES in mountainous areas.

## 4.3. Enhancing the Capacity of CES

Mapping CES could serve multiple purposes—from identifying hotspots for the tourism industry to planning land use. Incorporating the CES into land management and planning may be a measure to develop the local economy and protect natural resources. Benefits from ecological tourism are well known and a point of focus for the Chinese government [2]. The assessment of CES contributes

to guiding the tourism industry. While tourism has been developed in our study area, our results revealed that potential capacity still needed attention. Secondly, we found that some areas with high rarity did not present a high value of CES based on our approach. This is probably due to the low value of accessibility, which decreased the capacity of ecosystem services. Therefore, the roads must be considered in enhancing the capacity of services. On the contrary, some areas with a high rarity and accessibility should be limited to expand the area of the tourism industry so that these rare ecosystems could be conserved as well as maintain other ecosystem services (e.g., water retention, soil conservation) to support the capacity of CES.

Based on these findings, in order to improve and enhance the capacity of CES in the upper reaches of the Minjiang River, we suggest that local governments and decision makers should improve accessibility in these areas with high rarity and low accessibility. Considering ecological conservation and sustainable development, the area with a high rarity and accessibility should be restricted for enhancing the capacity of CES. Meanwhile, assessing ecological value may be useful to trade off among CES and other ecosystem services in areas with nonprovisioning services.

## 5. Conclusions

In this study, a new method incorporating the rarity and accessibility into was applied to assess CES in mountainous areas. The cumulative viewshed analysis quantified the accessibility of CES, which could identify the areas with nonprovisioning services. By K-means clustering algorithm, the distribution of rarity could be demonstrated and hotspots could be identified. The results showed that the area with low values of CES prevailed over the most area the upper reaches of the Minjiang River. Meanwhile, the areas with high CES value were mainly located in the southeastern part of the study area. In order to enhance the capacity of CES, we suggested that the areas with a high rarity and low accessibility should improve accessibility. On the contrary, the area with high rarity and accessibility should be intensively managed for ecological conservation. To sum up, our approach can be applied as a regional CES assessment model to identify hotspots for the tourism industry and can contribute to ecosystem services management.

**Author Contributions:** All authors were involved in the conceptualization of this study. Y.L. and Y.W. established the assessment model and formatted the figures. Q.L. and P.X. collected and analyzed the spatial data. The text was jointly written by all authors.

**Funding:** This research was funded by the West Light Foundation of Chinese Academy of Sciences, grant number Y7R280080 and National Nature Science Foundation of China, grant number 41701114.

**Acknowledgments:** On behalf of all contributors, we would like to give our sincere thanks to the editor of Sustainability, and to the reviewers for their constructive engagement with the manuscripts.

**Conflicts of Interest:** The authors declare no conflict of interest.

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
