# Peer review of "Incorporating Rarity and Accessibility Factors into the Cultural Ecosystem Services Assessment in Mountainous Areas: A Case Study in the Upper Reaches of the Minjiang River"

_sustainability, doi:10.3390/su11082203_

Round 1
Reviewer 1 Report
Cultural service is important but always be ignored among the four type ecosystem services. This paper focuses on the regional-scale cultural service and conduct quantitative analysis, which is good for the methodological research of cultural service assessment and mapping. However, the simple analysis mainly within the viewshed method. Meanwhile, the review part mentioned the gap about the ecosystem service demand, but not developed methods to fix it. Also it’s better to improve the English writing with minor revision.
Author Response
Dear Reviewers:
Thank you for your comments concerning our manuscript. Those comments are all valuable and very useful for revising and improving our paper. According to these comments and suggestions, we have revised the manuscript. Revised portion are marked in red in new manuscript.
Responds to the reviewers’s comments:
Responds to the reviewer’s comments:
Reviewer #1:
Cultural service is important but always be ignored among the four type ecosystem services. This paper focuses on the regional-scale cultural service and conduct quantitative analysis, which is good for the methodological research of cultural service assessment and mapping. However, the simple analysis mainly within the viewshed method. Meanwhile, the review part mentioned the gap about the ecosystem service demand, but not developed methods to fix it. Also it’s better to improve the English writing with minor revision.
Thanks reviewer for comments and hard work. The authors have revised the “the part of Introduction ” as follow.
The solution lies in the understanding of CES, which were produced by ecosystems as well as were influenced by social preferences in a region. Generally, changes of ecosystems determined the value of CES, while were related to human experience and appreciation of CES [15]. The previous studies suggested that landscape patches contributed to the value of CES, while assessment patches differ, including composition, structure and ecological maturity, could quantify landscape value [16]. In mountain areas, complex topography and geomorphology provided richness habitats, which shaped the rare landscape and local culture. Quantifying the effects of topography could identify the spatial differences of CES in terms of changes in topography in a landscape. Meanwhile, although the topography enhanced the capacity of CES, it influenced of accessibility on CES. Therefore, a reasonable approach should incorporate the accessibility and rarity into the assessment of CES.
In addition, the authors have improved and added some sentences in new manuscript (in red).

Reviewer 2 Report
The manuscript entitled " Incorporating Rarity and Accessibility Factors into the Cultural Ecosystem Services Assessment in Mountainous Areas: A Case Study in The Upper Reaches of the Minjiang River" shows a methodology of evaluation of cultural ecosystem services very useful for the sustainable development of tourism. It presents a first step in the sustainability of a mountainous area, although it would need more exhaustive recognition of the next steps, as it is based on the accessibility and rarity of ESCs, but what will happen to non-accessible areas, what will happen to those that are already accessible when roads need to be increased to encourage tourism and reach inaccessible areas as proposed in the article? A monitoring system over time should be proposed, as it is expected that the value of ESCs will change over time. I suggest the authors to read the following article in case it is of interest for the manuscript: MARTIN DE AGAR P., ORTEGA M., LOPEZ DE PABLO, C.T. 2016. A procedure of landscape services assessment based on mosaics of patches and boundaries JOURNAL OF ENVIRONMENTAL MANAGEMENT, 180:214-227
The manuscript is well written, except for the small details listed below.
Line 69, revise "the by cumulative".
Figure 3. Needs a more detailed explanation in the legend
Line 369 and 374 of the bibliography. Lack of numbering
Author Response
Dear Reviewers:
Thank you for your comments concerning our manuscript. Those comments are all valuable and very useful for revising and improving our paper. According to these comments and suggestions, we have revised the manuscript. Revised portion are marked in red in new manuscript.
Responds to the reviewer’s comments:
Reviewer #2:
The manuscript entitled " Incorporating Rarity and Accessibility Factors into the Cultural Ecosystem Services Assessment in Mountainous Areas: A Case Study in The Upper Reaches of the Minjiang River" shows a methodology of evaluation of cultural ecosystem services very useful for the sustainable development of tourism. It presents a first step in the sustainability of a mountainous area, although it would need more exhaustive recognition of the next steps, as it is based on the accessibility and rarity of ESCs, but what will happen to non-accessible areas, what will happen to those that are already accessible when roads need to be increased to encourage tourism and reach inaccessible areas as proposed in the article?
In our study, the authors focus on the distribution of cultural ecosystem services in current conditions (natural resources, infrastructure). There may be high capacity of cultural ecosystem services in non-accessible areas where it could not provide services for human because of nature reserves for biodiversity and complex topography. However, the other ecosystem services, including water retention, soil conservation, could provide in these non-accessible areas. In the future, enhanced the capacity of cultural ecosystem services would be mostly concentrated in accessible areas. Meanwhile, the authors added some sentences in the part of discussion.
A monitoring system over time should be proposed, as it is expected that the value of ESCs will change over time. I suggest the authors to read the following article in case it is of interest for the manuscript: MARTIN DE AGAR P., ORTEGA M., LOPEZ DE PABLO, C.T. 2016. A procedure of landscape services assessment based on mosaics of patches and boundaries JOURNAL OF ENVIRONMENTAL MANAGEMENT, 180:214-227.
Thanks reviewer for suggestion. The authors have revised the manuscript and cited the article in this study as follow.
The solution lies in the understanding of CES, which were produced by ecosystems as well as were influenced by social preferences in a region. Generally, changes of ecosystems determined the value of CES, while were related to human experience and appreciation of CES [15]. The previous studies suggested that landscape patches contributed to the value of CES, while assessment patches differ, including composition, structure and ecological maturity, could quantify landscape value [16]. In mountain areas, complex topography and geomorphology provided richness habitats, which shaped the rare landscape and local culture. Quantifying the effects of topography could identify the spatial differences of CES in terms of changes in topography in a landscape. Meanwhile, although the topography enhanced the capacity of CES, it influenced of accessibility on CES. Therefore, a reasonable approach should incorporate the accessibility and rarity into the assessment of CES.
The manuscript is well written, except for the small details listed below.
Line 69, revise "the by cumulative".
The authors have revised in new manuscript as follow.
(1) assessing the accessibility by cumulative viewshed analysis algorithm,
Figure 3. Needs a more detailed explanation in the legend
The authors have revised in new manuscript as follow.
The lower values are calculated by SSE and Silhouette coefficient, the better the clustering quality of k mean algorithm is.
Line 369 and 374 of the bibliography. Lack of numbering
The authors have revised in new manuscript as follow.
21. Weiss, A. (2001). Topographic position and landforms analysis. Poster presentation, ESRI user conference, San Diego, CA.
22. Lletı, R.;Ortiz, M. C.;Sarabia, L. A.Sánchez, M. S., Selecting variables for k-means cluster analysis by using a genetic algorithm that optimises the silhouettes. Analytica Chimica Acta. 2004, 515(1): 87-100.
23. Su, T.Dy, J. (2004). A deterministic method for initializing k-means clustering. Tools with Artificial Intelligence, 2004. ICTAI 2004. 16th IEEE International Conference on, IEEE.
